## [Peer Review File · EMBO Reports]

EZH2 directs HER2+ breast cancer progression through the modulation of epithelial plasticity

Linshan Liu, Ellie Massey, Dongmei Zuo, Alain Pacis, Mert Demirdizen, Elizabeth Podleszanski, Jessica Cinkornpumin, Yu Gu, Hailey Proud, Virginie Sanguin-Gendreau, Vasilios Papavasiliou, Ishtiaque Hossain, Zhengze Jiang, Harvey Smith, William Pastor, Paolo Ceppi, and William Muller

Corresponding author(s): William Muller (william.muller@mcgill.ca)

Review Timeline:

Submission Date:	15th Jul 25
Editorial Decision:	13th Aug 25
Revision Received:	24th Oct 25
Editorial Decision:	5th Dec 25
Revision Received:	11th Dec 25
Accepted:	7th Jan 26

Editor: Achim Breiling

Transaction Report:

Dear Dr. Muller,

Thank you for the transfer of your manuscript to EMBO reports. I have now received reports from the three referees that were asked to evaluate your study, which can be found at the end of this email. As you will see, the referees think that these findings are of high interest. However, they have several comments, concerns, and suggestions, indicating that a major revision of the manuscript is necessary to allow publication of the study in EMBO reports. As the reports are below, and all the referee concerns need to be addressed, I will not detail them here.

Given the constructive referee comments, I would thus like to invite you to revise your manuscript with the understanding that the concerns of the referees must be addressed in the revised manuscript and/or in a detailed point-by-point response. Acceptance of your manuscript will depend on a positive outcome of a second round of review. It is EMBO reports policy to allow a single round of revision only and acceptance of the manuscript will therefore depend on the completeness of your responses included in the next, final version of the manuscript.

1) a .docx formatted version of the final manuscript text (including legends for main figures, EV figures and tables), but without the figures included. Figure legends should be compiled at the end of the manuscript text.

2) individual production quality figure files as .eps, .tif, .jpg (one file per figure), of main figures and EV figures. Please upload these as separate, individual files upon re-submission.

4) a complete author checklist, which you can download from our author guidelines

(<https://www.embopress.org/page/journal/14693178/authorguide>). Please insert page numbers in the checklist to indicate where the requested information can be found in the manuscript. The completed author checklist will also be part of the RPF.

5) that primary datasets produced in this study (e.g. RNA-seq, ChIP-seq, structural and array data) are deposited in an appropriate public database. If no primary datasets have been deposited, please also state this in a dedicated section (e.g. 'No

primary datasets have been generated and deposited'), see below.

The accession numbers and database should be listed in a formal "Data Availability" section that follows the model below. This is now mandatory (like the COI statement). Please note that the Data Availability Section is restricted to new primary data that are part of this study. This section is mandatory. As indicated above, if no primary datasets have been deposited, please state this in this section

Data availability

6) We now request the publication of original source data with the aim of making primary data more accessible and transparent to the reader. You will receive a separate email with instructions for providing source data with your revised manuscript, including information how to upload and organize the files.

8) Regarding data quantification and statistics, please make sure that the number "n" for how many independent experiments were performed, their nature (biological versus technical replicates), the bars and error bars (e.g. SEM, SD) and the test used to calculate p-values is indicated in the respective figure legends (also for EV and Appendix figures). Please also check that all the p-values are explained in the legend, and that these fit to those shown in the figure. Please provide statistical testing where applicable. Please avoid the phrase 'independent experiment', but clearly state if these were biological or technical replicates. Please also indicate (e.g. with n.s.) if testing was performed, but the differences are not significant. In case n=2, please show the data as separate datapoints without error bars and statistics. See also: <http://www.embopress.org/page/journal/14693178/authorguide#statisticalanalysis>

9) Please add scale bars of similar style and thickness to all microscopic images, using clearly visible black or white bars (depending on the background). Please place these in the lower right corner of the images themselves. Please do not write on or near the bars in the image but define the size in the respective figure legend.

10) Please also note our reference format:
<http://www.embopress.org/page/journal/14693178/authorguide#referencesformat>

12) We now use CRediT to specify the contributions of each author in the journal submission system. CRediT replaces the author contribution section. Please use the free text box to provide more detailed descriptions and do NOT provide your final manuscript text file with an author contributions section. See also our guide to authors: <https://www.embopress.org/page/journal/14693178/authorguide#authorshipguidelines>

13) All Materials and Methods need to be described in the main text using our 'Structured Methods' format, which is required for all research articles. According to this format, the Methods section should include a Reagents and Tools Table (listing key

reagents, experimental models, software, and relevant equipment and including their sources and relevant identifiers), uploaded as separate file, and a Methods section in which we encourage the authors to describe their methods using a step-by-step protocol format with bullet points, to facilitate the adoption of the methodologies across labs. More information on how to adhere to this format as well as downloadable templates (.doc) for the Reagents and Tools Table can be found in our author guidelines (section 'Structured Methods'):

14) Please add up to 5 keywords to the manuscript and order the manuscript sections like this, using only these names: Title page - Abstract - Keywords - Introduction - Results - Discussion - Methods - Data availability section - Acknowledgements (please put here all the funding information) - Disclosure and Competing Interests Statement - References - Figure legends - Expanded View Figure legends

15) Please make sure that all the funding information is also entered into the online submission system and that it is complete and similar to the one in the acknowledgement section of the manuscript text file.

I look forward to seeing a revised form of your manuscript when it is ready.

Yours sincerely,

Referee #1:

Liu and colleagues present a compelling study that addresses a relevant and timely question in the field of breast cancer research, particularly regarding aggressive HER2-positive tumors, which account for approximately 15-20% of diagnosed cases and frequently develop resistance to standard therapies. Using elegant and well-designed experiments in genetically engineered mouse models, the authors investigate the role of EZH2 in tumor progression. Although the link between EZH2 and breast cancer has been previously reported (reviews: PMID: 38670045, PMID: 38331087, PMID: 34335938) their findings demonstrate that EZH2 promotes both tumor initiation and metastasis. Notably, EZH2 depletion results in a reduction of the basal cell population and a shift toward a more luminal identity, as indicated by increased expression of the estrogen receptor (ESR1). Importantly, this upregulation of ESR1 confers sensitivity to tamoxifen. Collectively, these findings suggest that targeting EZH2 may offer a promising therapeutic approach for managing HER2-positive breast cancers that are resistant to conventional treatments.

Major points:

1-That said, a key mechanistic point remains insufficiently addressed. To convincingly demonstrate that the reduction of H3K27me3 at the ESR1 locus is directly mediated by EZH2 (Figure 6H), the authors should perform EZH2 ChIP-qPCR at the ESR1 gene, along with control loci whose H3K27me3 levels remain stable in both EZH2-expressing and EZH2-depleted conditions. In addition, similar ChIP-qPCR experiments targeting EMT regulators and basal lineage markers would help determine whether these genes are also directly regulated by EZH2, thereby further substantiating the proposed mechanism.

2-From a mechanistic perspective, the manuscript presents a substantial number of genes that are downregulated upon EZH2 depletion. Given that EZH2 is a core component of the repressive PRC2 complex, how can this downregulation be explained? It would be important for the authors to consider whether the 420 genes found to be downregulated upon EZH2 loss (Figure 3A-B) might be controlled through a non-canonical mechanism, in which EZH2, independently of PRC2, promotes gene expression via interactions with transcription factors or oncogenic partners, as previously described (doi: 10.3389/fonc.2023.1233953). If this is the case, the model presented in Figure 7 may need to be revised accordingly.

As a minor point, throughout the manuscript, the presentation of bar graphs -used to quantify Western blots, immunofluorescence, metastasis, migration, etc.- would benefit from larger lettering to improve readability. In Figure 5E, the labeling indicating which parts of the graph correspond to Notch3 and ESR1 could be enlarged to improve clarity and readability.

Referee #2:

Evaluation of EMBOR-2025-62337V1-T manuscript "EZH2 directs breast cancer progression through the modulation of epithelial cellular plasticity"

By authors: William Muller, Ellie Massey, Linshan Liu, Dongmei Zuo, Alain Pacis, Mert Demirdizen, Jessica Cinkornpumin, Yu Gu, Elizabeth Podleszanski, Hailey Proud, Virginie Sanguin-Gendreau, Vasilios Papavasiliou, Zhengze Jiang, Harvey Smith, William Pastor, and Paolo Ceppi

This is a good story with strong experimental support. In the story authors present evidence, both in vivo and in vitro, that HER2 aggressiveness in mammary cancer, especially in terms of invasiveness, is likely to be at least partially mediated by EZH2. Furthermore the results suggest that combination of ER inhibitor tamoxifen and EZH2 inhibitors could be beneficial for HER2+ breast cancer. The study is rather complete and I do not have a lot to comment.

Critical comments:

1. Western blots especially in Figures 2B,2 C and 4D are cropped too much and they miss molecular weight markers. The quantification suffers if the bands are not properly included. It is also difficult to see if the presented ones are the major bands due to this and due to the lack of molecular weight markers.
2. Figures 2A and B would need to be clarified so that the gene names would also be with the corresponding graph. Now we must assume that the graphs are in the order of the gene names presented in the figure text. It would be better if the reader did not need to assume anything.
3. The title is a bit too elusive. Perhaps something about the role of EZH2 in the invasiveness of HER+ positive breast cancer would be more close to the observations.

Referee #3:

Linshan Liu et al. analyses the role of EZH2 in HER2+ positive breast cancer, both in vivo and in vitro systems. They described that lack of function of EZH2 in this breast cancer subtype delays tumor formation and significantly reduces metastatic dissemination potential. The authors attribute these phenotypes to changes in cell states derived from the lack of PRC2 repression, leading to the acquisition of an epithelial identity and a concomitant loss of mesenchymal features. In addition, EZH2 loss leads to the upregulation of Estrogen receptor alpha (Esr1), which emerges as a potential therapeutical approach against breast cancer cells due to increased sensitivity to endocrine therapies, although these last results are only validated in vitro.

Overall, the manuscript is well written, and the experiments appear technically robust and thoughtfully designed. However, I am concerned about the conclusions drawn by the authors, as there is existing literature that points in a different direction regarding the role of EZH2 regulating epithelial to mesenchymal plasticity in breast cancer cells. This conflicting evidence has not been addressed or discussed in the current version of the manuscript. I further detail this and other points below.

1. There are several papers showing strong evidence that EZH2 repress mesenchymal genes in different types of breast cancer. Thus, lack of EZH2 results in acquisition of a mesenchymal identity. This loss of epithelial-mesenchymal plasticity also impacts on the dissemination capacity of the cancer cells (Hirukawa, Smith et al. 2018, Zhang, Donaher et al. 2022, Gallardo, López-Onieva et al. 2024).

One possibility is that EZH2 function is highly context-dependent, and in this model of HER2-induced tumorigenesis, the lack of function of EZH2 mainly affects cell cycle regulation (i.e CDKN2A locus remains intact and is described as a PRC2 target in different cell types). The absence of additional oncogenic mutations may prevent the acquisition of a mesenchymal phenotype (Serresi, Gargiulo et al. 2016).

However, I wonder whether this discrepancy could instead be due to the remarkably compensation of EZH1 in this model. Although a global reduction of H3K27me3 can be observed in the experiments presented in Figures 1,2 and S4, the reduction is modest compared to the pronounced depletion of H3K27me3 observed upon treatment with EZH2 inhibitors (figure 6F). Moreover, H3K27me3 levels seem to be partially restored when comparing time point 12w (Fig. 1D) with the endpoint (Fig. 2A). In addition, H3K27me3 ChIP-seq data shows that there are at least 2000 genes with high enrichment of H3K27me at their promoter region (Fig 8H), but only the 40% of these potential targets present a significant change at mRNA level (Fig.3B). Thus, I would like to know the opinion of the authors about these works and how they can explain the difference of their results.

2.To clarify the role of EZH2 in epithelial-mesenchymal plasticity, I think it would be interesting to analyze how HER2+ cells respond to a more complete depletion of H3K27me3. Specifically, the derived cell lines shown in Fig.2D could be treated with EZH2 inhibitors EPZ6438 or GSK126, due to although these inhibitors display higher affinity for EZH2, they are able to inhibit EZH1 too. The authors could check by western blot if the EZH2i treatment results in a higher reduction of H3K27me3. RNA-seq would provide a comprehensive readout of the experiment to know if cells under a complete loss of H3K27me3 preferentially up or downregulate the EMT program. Western blot of E-Cadherin, Vimentin or Snail could complement these findings, although the EMT markers are known to vary by cell type. Migration assay would be also informative to translate the gene expression changes into functionality features.

3.Related to point 2, I suggest the authors explore how EPZ6438 and GSK126 affect epithelial-mesenchymal dynamics in additional HER2+ breast cancer cell lines such as SKBR3, HCC1954, and JIMT-1. While RNA-seq would be ideal, a targeted RT-qPCR panel of epithelial and mesenchymal markers (identified based on prior transcriptomic profiling) could provide a cost-effective alternative.

4.In Fig. 2D-G, the authors show that Ezh2-null cells form fewer tumors in mammary glands and lungs, and attribute this to reduced epithelial-mesenchymal plasticity. However, it would be helpful to determine whether these cells also exhibit reduced proliferation in vitro, which could independently contribute to the observed in vivo phenotype.

5.The idea of using EZH2 inhibition to sensitize HER2+ breast cancer cells to endocrine therapy is compelling. However, this effect is only shown in SKBR3 cells. The manuscript would benefit from additional validation. The authors would like to test in vivo, if the administration of the endocrine therapy in the MMTV-rtTA/AEIC system is more effective in the EZH2fl/fl than the EZH2wt/wt background. At minimum, it would be valuable to test in the derived cell lines (Fig.2D) if ERa is induced in the Ezh2 null cells and they show higher sensitivity to Tamoxifen. Additionally, tamoxifen sensitivity assays in HCC1954 and JIMT-1 cells treated with EZH2 inhibitors would also strengthen the translational potential of these findings.

Other minor comments that I think would improve the message of the manuscript are:

1.In line 465, the authors described H3K27me3 levels in Ezh2 null mammary glands as "residual". In my opinion, this may be misleading, as the data suggest that approximately one-third of H3K27me3 remains unaffected. The term "remaining" or "partial" might be more accurate than "residual".

2.The current title is quite attractive and suggests an important idea. However, I think it suggests a broad conclusion that may not be fully supported outside the HER2 context. I recommend specifying the subtype, e.g: "EZH2 directs HER2-enriched breast cancer progression through the modulation of epithelial cellular plasticity"

References:

- Gallardo, A., L. López-Onieva, E. Belmonte-Reche, I. Fernández-Rengel, A. Serrano-Prados, A. Molina, A. Sánchez-Pozo and D. Landeira (2024). "EZH2 represses mesenchymal genes and upholds the epithelial state of breast carcinoma cells." *Cell Death & Disease* 15(8): 609.
- Hirukawa, A., H. W. Smith, D. Zuo, C. R. Dufour, P. Savage, N. Bertos, R. M. Johnson, T. Bui, G. Bourque, M. Basik, V. Giguère, M. Park and W. J. Muller (2018). "Targeting EZH2 reactivates a breast cancer subtype-specific anti-metastatic transcriptional program." *Nature Communications* 9(1): 2547.
- Serresi, M., G. Gargiulo, N. Proost, B. Siteur, M. Cesaroni, M. Koppens, H. Xie, K. D. Sutherland, D. Hulsman, E. Citterio, S. Orkin, A. Berns and M. van Lohuizen (2016). "Polycomb Repressive Complex 2 Is a Barrier to KRAS-Driven Inflammation and Epithelial-Mesenchymal Transition in Non-Small-Cell Lung Cancer." *Cancer Cell* 29(1): 17-31.
- Zhang, Y., J. L. Donaher, S. Das, X. Li, F. Reinhardt, J. A. Krall, A. W. Lambert, P. Thiru, H. R. Keys, M. Khan, M. Hofree, M. M. Wilson, O. Yedier-Bayram, N. A. Lack, T. T. Onder, T. Bagci-Onder, M. Tyler, I. Tirosh, A. Regev, J. A. Lees and R. A. Weinberg (2022). "Genome-wide CRISPR screen identifies PRC2 and KMT2D-COMPASS as regulators of distinct EMT trajectories that contribute differentially to metastasis." *Nature Cell Biology* 24(4): 554-564.

William J. Muller
Professor of Biochemistry
Rosalind and Morris Goodman Cancer Centre
McGill University
1160 Pine Avenue West
Montreal, QC, Canada H3A 1A3

McGill

24th October 2025

Dear reviewers,

We sincerely thank all three Reviewers for your insightful comments and feedback on our manuscript entitled '**EZH2 directs HER2+ breast cancer progression through the modulation of epithelial plasticity**'. Here, we give a point-by-point clarification on each of your comments and outline how we have addressed each of your concerns in our revised manuscript.

REVIEWERS' COMMENTS

REFEREE #1:

Liu and colleagues present a compelling study that addresses a relevant and timely question in the field of breast cancer research, particularly regarding aggressive HER2-positive tumors, which account for approximately 15-20% of diagnosed cases and frequently develop resistance to standard therapies. Using elegant and well-designed experiments in genetically engineered mouse models, the authors investigate the role of EZH2 in tumor progression. Although the link between EZH2 and breast cancer has been previously reported (reviews: PMID: 38670045, PMID: 38331087, PMID: 34335938) their findings demonstrate that EZH2 promotes both tumor initiation and metastasis. Notably, EZH2 depletion results in a reduction of the basal cell population and a shift toward a more luminal identity, as indicated by increased expression of the estrogen receptor (ESR1). Importantly, this upregulation of ESR1 confers sensitivity to tamoxifen. Collectively, these findings suggest that targeting EZH2 may offer a promising therapeutic approach for managing HER2-positive breast cancers that are resistant to conventional treatments.

Major points:

1-That said, a key mechanistic point remains insufficiently addressed. To convincingly demonstrate that the reduction of H3K27me3 at the ESR1 locus is directly mediated by EZH2 (Figure 6H), the authors should perform EZH2 ChIP-qPCR at the ESR1 gene, along with control loci whose H3K27me3 levels remain stable in both EZH2-expressing and EZH2-depleted conditions. In addition, similar ChIP-qPCR experiments targeting EMT regulators and basal lineage markers would help determine whether these genes are also directly regulated by EZH2, thereby further substantiating the proposed mechanism.

2-From a mechanistic perspective, the manuscript presents a substantial number of genes that are downregulated upon EZH2 depletion. Given that EZH2 is a core component of the repressive PRC2 complex, how can this downregulation be explained? It would be important for the authors to consider whether the 420 genes found to be downregulated upon EZH2 loss (Figure 3A-B) might be controlled through a non-canonical mechanism, in which EZH2, independently of PRC2, promotes gene expression via interactions with transcription factors or oncogenic partners, as previously described (doi: 10.3389/fonc.2023.1233953). If this is the case, the model presented in Figure 7 may need to be revised accordingly.

William J. Muller
Professor of Biochemistry
Rosalind and Morris Goodman Cancer Centre
McGill University
1160 Pine avenue West
Montreal, QC, Canada H3A 1A3

McGill

As a minor point, throughout the manuscript, the presentation of bar graphs -used to quantify Western blots, immunofluorescence, metastasis, migration, etc.- would benefit from larger lettering to improve readability.

In Figure 5E, the labeling indicating which parts of the graph correspond to Notch3 and ESR1 could be enlarged to improve clarity and readability.

REFEREE #2:

Evaluation of EMBOR-2025-62337V1-T manuscript "EZH2 directs breast cancer progression through the modulation of epithelial cellular plasticity"

By authors: William Muller, Ellie Massey, Linshan Liu, Dongmei Zuo, Alain Pacis, Mert Demirdizen, Jessica Cinkornpumin, Yu Gu, Elizabeth Podleszanski, Hailey Proud, Virginie Sanguin-Gendreau, Vasilios Papavasiliou, Zhengze Jiang, Harvey Smith, William Pastor, and Paolo Ceppi

This is a good story with strong experimental support. In the story authors present evidence, both in vivo and in vitro, that HER2 aggressiveness in mammary cancer, especially in terms of invasiveness, is likely to be at least partially mediated by EZH2. Furthermore the results suggest that combination of ER inhibitor tamoxifen and EZH2 inhibitors could be beneficial for HER2+ breast cancer. The study is rather complete and I do not have a lot to comment.

Critical comments:

1. Western blots especially in Figures 2B, 2 C and 4D are cropped too much and they miss molecular weight markers. The quantification suffers if the bands are not properly included. It is also difficult to see if the presented ones are the major bands due to this and due to the lack of molecular weight markers.
2. Figures 2A and B would need to be clarified so that the gene names would also be with the corresponding graph. Now we must assume that the graphs are in the order of the gene names presented in the figure text. It would be better if the reader did not need to assume anything.
3. The title is a bit too elusive. Perhaps something about the role of EZH2 in the invasiveness of HER+ positive breast cancer would be more close to the observations.

REFEREE #3:

Linshan Liu et al. analyses the role of EZH2 in HER2+ positive breast cancer, both in vivo and in vitro systems. They described that lack of function of EZH2 in this breast cancer subtype delays tumor formation and significantly reduces metastatic dissemination potential. The authors attribute these phenotypes to changes in cell states derived from the lack of PRC2 repression, leading to the acquisition of an epithelial identity and a concomitant loss of mesenchymal features. In addition, EZH2 loss leads to the upregulation of Estrogen receptor alpha (Esr1), which emerges as a potential therapeutical approach against breast cancer cells

William J. Muller
Professor of Biochemistry
Rosalind and Morris Goodman Cancer Centre
McGill University
1160 Pine avenue West
Montreal, QC, Canada H3A 1A3

McGill

due to increased sensitivity to endocrine therapies, although these last results are only validated in vitro.

Overall, the manuscript is well written, and the experiments appear technically robust and thoughtfully designed. However, I am concerned about the conclusions drawn by the authors, as there is existing literature that points in a different direction regarding the role of EZH2 regulating epithelial to mesenchymal plasticity in breast cancer cells. This conflicting evidence has not been addressed or discussed in the current version of the manuscript. I further detail this and other points below.

1. There are several papers showing strong evidence that EZH2 repress mesenchymal genes in different types of breast cancer. Thus, lack of EZH2 results in acquisition of a mesenchymal identity. This loss of epithelial-mesenchymal plasticity also impacts on the dissemination capacity of the cancer cells (Hirukawa, Smith et al. 2018, Zhang, Donaher et al. 2022, Gallardo, López-Onieva et al. 2024).

One possibility is that EZH2 function is highly context-dependent, and in this model of HER2-induced tumorigenesis, the lack of function of EZH2 mainly affects cell cycle regulation (i.e CDKN2A locus remains intact and is described as a PRC2 target in different cell types). The absence of additional oncogenic mutations may prevent the acquisition of a mesenchymal phenotype (Serresi, Gargiulo et al. 2016).

However, I wonder whether this discrepancy could instead be due to the remarkably compensation of EZH1 in this model. Although a global reduction of H3K27me3 can be observed in the experiments presented in Figures 1,2 and S4, the reduction is modest compared to the pronounced depletion of H3K27me3 observed upon treatment with EZH2 inhibitors (figure 6F). Moreover, H3K27me3 levels seem to be partially restored when comparing time point 12w (Fig. 1D) with the endpoint (Fig. 2A). In addition, H3K27me3 ChIP-seq data shows that there are at least 2000 genes with high enrichment of H3K27me at their promoter region (Fig 8H), but only the 40% of these potential targets present a significant change at mRNA level (Fig.3B). Thus, I would like to know the opinion of the authors about these works and how they can explain the difference of their results.

2. To clarify the role of EZH2 in epithelial-mesenchymal plasticity, I think it would be interesting to analyze how HER2+ cells respond to a more complete depletion of H3K27me3. Specifically, the derived cell lines shown in Fig.2D could be treated with EZH2 inhibitors EPZ6438 or GSK126, due to although these inhibitors display higher affinity for EZH2, they are able to inhibit EZH1 too. The authors could check by western blot if the EZH2i treatment results in a higher reduction of H3K27me3. RNA-seq would provide a comprehensive readout of the experiment to know if cells under a complete loss of H3K27me3 preferentially up or downregulate the EMT program. Western blot of E-Cadherin, Vimentin or Snail could complement these findings, although the EMT markers are known to vary by cell type. Migration assay would be also informative to translate the gene expression changes into functionality features.

3. Related to point 2, I suggest the authors explore how EPZ6438 and GSK126 affect

epithelial-mesenchymal dynamics in additional HER2+ breast cancer cell lines such as SKBR3, HCC1954, and JIMT-1. While RNA-seq would be ideal, a targeted RT-qPCR panel of epithelial and mesenchymal markers (identified based on prior transcriptomic profiling) could provide a cost-effective alternative.

4. In Fig. 2D-G, the authors show that Ezh2-null cells form fewer tumors in mammary glands and lungs, and attribute this to reduced epithelial-mesenchymal plasticity. However, it would be helpful to determine whether these cells also exhibit reduced proliferation *in vitro*, which could independently contribute to the observed *in vivo* phenotype.

5. The idea of using EZH2 inhibition to sensitize HER2+ breast cancer cells to endocrine therapy is compelling. However, this effect is only shown in SKBR3 cells. The manuscript would benefit from additional validation. The authors would like to test *in vivo*, if the administration of the endocrine therapy in the MMTV-rtTA/AEIC system is more effective in the EZH2^{fl/fl} than the EZH2^{wt/wt} background. At minimum, it would be valuable to test in the derived cell lines (Fig. 2D) if ERα is induced in the Ezh2 null cells and they show higher sensitivity to Tamoxifen. Additionally, tamoxifen sensitivity assays in HCC1954 and JIMT-1 cells treated with EZH2 inhibitors would also strengthen the translational potential of these findings.

Other minor comments that I think would improve the message of the manuscript are:

1. In line 465, the authors described H3K27me3 levels in Ezh2 null mammary glands as "residual". In my opinion, this may be misleading, as the data suggest that approximately one-third of H3K27me3 remains unaffected. The term "remaining" or "partial" might be more accurate than "residual".

2. The current title is quite attractive and suggests an important idea. However, I think it suggests a broad conclusion that may not be fully supported outside the HER2 context. I recommend specifying the subtype, e.g: "EZH2 directs HER2-enriched breast cancer progression through the modulation of epithelial cellular plasticity"

References:

- Gallardo, A., L. López-Onieva, E. Belmonte-Reche, I. Fernández-Rengel, A. Serrano-Prados, A. Molina, A. Sánchez-Pozo and D. Landeira (2024). "EZH2 represses mesenchymal genes and upholds the epithelial state of breast carcinoma cells." *Cell Death & Disease* 15(8): 609.
- Hirukawa, A., H. W. Smith, D. Zuo, C. R. Dufour, P. Savage, N. Bertos, R. M. Johnson, T. Bui, G. Bourque, M. Basik, V. Giguère, M. Park and W. J. Muller (2018). "Targeting EZH2 reactivates a breast cancer subtype-specific anti-metastatic transcriptional program." *Nature Communications* 9(1): 2547.
- Serresi, M., G. Gargiulo, N. Proost, B. Siteur, M. Cesaroni, M. Koppens, H. Xie, K. D. Sutherland, D. Hulsman, E. Citterio, S. Orkin, A. Berns and M. van Lohuizen (2016). "Polycomb Repressive Complex 2 Is a Barrier to KRAS-Driven Inflammation and Epithelial-Mesenchymal Transition in Non-Small-Cell Lung Cancer." *Cancer Cell* 29(1): 17-31.
- Zhang, Y., J. L. Donaher, S. Das, X. Li, F. Reinhardt, J. A. Krall, A. W. Lambert, P. Thiru, H. R. Keys, M. Khan, M. Hofree, M. M. Wilson, O. Yedier-Bayram, N. A. Lack, T. T. Onder, T. Bagci-Onder, M. Tyler, I. Tirosh, A. Regev, J. A. Lees and R. A. Weinberg (2022). "Genome-

William J. Muller
Professor of Biochemistry
Rosalind and Morris Goodman Cancer Centre
McGill University
1160 Pine avenue West
Montreal, QC, Canada H3A 1A3

McGill

wide CRISPR screen identifies PRC2 and KMT2D-COMPASS as regulators of distinct EMT trajectories that contribute differentially to metastasis." *Nature Cell Biology* 24(4): 554-564.

REVIEWERS' COMMENTS: Point by point response

Reviewer #1

A key mechanistic point remains insufficiently addressed. To convincingly demonstrate that the reduction of H3K27me3 at the ESR1 locus is directly mediated by EZH2 (Figure 6H), the authors should perform EZH2 ChIP-qPCR at the ESR1 gene, along with control loci whose H3K27me3 levels remain stable in both EZH2-expressing and EZH2-depleted conditions. In addition, similar ChIP-qPCR experiments targeting EMT regulators and basal lineage markers would help determine whether these genes are also directly regulated by EZH2, thereby further substantiating the proposed mechanism.

To directly address Reviewer #1's comment, we performed EZH2 ChIP-qPCR in SK-BR-3 parental and lapatinib resistant cells at the *ESR1* gene, along with control loci (*RPL30* and *GAPDH*) where H3K27me3 enrichment is not found. Primers were designed using our H3K27me3 ChIP-Sequencing data and are indicated in Appendix Table S5. As shown in revised Fig. 6I, we convincingly demonstrate that *ESR1* is enriched in comparison to these control loci, indicating that EZH2 is directly bound at the *ESR1* locus. We further validated this in JIMT-1 cells (see attached figure).

In addition, we also probed our H3K27me3 ChIP-Sequencing data to find other luminal markers which could be directly regulated by EZH2. The results showed that the genes encoding luminal markers (*FOXA1*, *GATA3*, *NOTCH3*, *EPCAM*, *CDH1*, *KRT8*) exhibit little H3K27me3 enrichment (see below). Taken together, these data argue that they are indirectly upregulated as a product wider ER α dependent luminal differentiation program (Bernardo and Keri, 2012; Eeckhoutte J *et al.*, 2007; Wilson and Giguère, 2012).

From a mechanistic perspective,

William J. Muller
Professor of Biochemistry
Rosalind and Morris Goodman Cancer Centre
McGill University
1160 Pine avenue West
Montreal, QC, Canada H3A 1A3

McGill

the manuscript presents a substantial number of genes that are downregulated upon EZH2 depletion. Given that EZH2 is a core component of the repressive PRC2 complex, how can this downregulation be explained? It would be important for the authors to consider whether the 420 genes found to be downregulated upon EZH2 loss (Figure 3A-B) might be controlled through a non-canonical mechanism, in which EZH2, independently of PRC2, promotes gene expression via interactions with transcription factors or oncogenic partners, as previously described (doi: 10.3389/fonc.2023.1233953). If this is the case, the model presented in Figure 7 may need to be revised accordingly.

We agree with the reviewers' comments that a substantial number of genes were downregulated upon EZH2 loss, although this is consistent with other conditional EZH2 loss studies (Hirukawa *et al.*, 2018; Liu *et al.*, 2023). These downregulated genes are likely controlled through non-canonical EZH2 functions, and we have adjusted the text in the manuscript to better acknowledge EZH2 non-canonical functions (Zimmerman *et al.*, 2023). Further, we have revised the model in Figure 7 to better reflect our model of Ezh2 loss. **As a minor point, throughout the manuscript, the presentation of bar graphs -used to quantify Western blots, immunofluorescence, metastasis, migration, etc.- would benefit from larger lettering to improve readability.**

We acknowledge that some bar graphs are hard to read and therefore we have edited the graphs to have larger font where possible to improve readability.

In Figure 5E, the labeling indicating which parts of the graph correspond to Notch3 and ESR1 could be enlarged to improve clarity and readability.

As suggested, we have enlarged the labels of *Esr1* and *Notch3* on Figure 5E to improve clarity.

Reviewer #2

Western blots especially in Figures 2B,2C and 4D are cropped too much and they miss molecular weight markers. The quantification suffers if the bands are not properly included. It is also difficult to see if the presented ones are the major bands due to this and due to the lack of molecular weight markers.

Due to lack of space, we have cropped western blot images as concisely as possible in the Figure to show the entire protein band. We can assure reviewer #2 that quantifications were performed using the full blot image in ImageStudio (LICORbio, v6.1), ensuring that the entire protein band was quantified accurately, and we have adjusted the Materials and Methods to reflect this. We have additionally provided the molecular weight of the protein shown to the right of each image, as can now be seen in Figure 2B, 2C and 4D, and other western blot Figures. Full blots with the protein ladder and labelled molecular weights are also provided in the source data.

Figures 2A and B would need to be clarified so that the gene names would also be with the corresponding graph. Now we must assume that the graphs are in the order of the

William J. Muller
Professor of Biochemistry
Rosalind and Morris Goodman Cancer Centre
McGill University
1160 Pine avenue West
Montreal, QC, Canada H3A 1A3

McGill

gene names presented in the figure text. It would be better if the reader did not need to assume anything.

To address this minor comment, in the revised manuscript we highlighted what is being quantified on the y axis in Figure 2A (e.g. HER2+Ezh2+ cells). For western blot quantifications such as Figure 2B, we highlight what protein is being quantified in the graph title. We have enlarged graphs where possible to improve font sizes of these titles.

The title is a bit too elusive. Perhaps something about the role of EZH2 in the invasiveness of HER+ positive breast cancer would be more close to the observations.

We have adjusted the title to reflect that our findings are specifically in HER2-enriched breast cancer.

Reviewer #3

There are several papers showing strong evidence that EZH2 repress mesenchymal genes in different types of breast cancer. Thus, lack of EZH2 results in acquisition of a mesenchymal identity. This loss of epithelial-mesenchymal plasticity also impacts on the dissemination capacity of the cancer cells (Hirukawa, Smith et al. 2018, Zhang, Donaher et al. 2022, Gallardo, López-Onieva et al. 2024). One possibility is that EZH2 function is highly context-dependent, and in this model of HER2-induced tumorigenesis, the lack of function of EZH2 mainly affects cell cycle regulation (i.e CDKN2A locus remains intact and is described as a PRC2 target in different cell types). The absence of additional oncogenic mutations may prevent the acquisition of a mesenchymal phenotype (Serresi, Gargiulo et al. 2016). However, I wonder whether this discrepancy could instead be due to the remarkably compensation of EZH1 in this model. Although a global reduction of H3K27me3 can be observed in the experiments presented in Figures 1,2 and S4, the reduction is modest compared to the pronounced depletion of H3K27me3 observed upon treatment with EZH2 inhibitors (figure 6F). Moreover, H3K27me3 levels seem to be partially restored when comparing time point 12w (Fig. 1D) with the endpoint (Fig. 2A). In addition, H3K27me3 ChIP-seq data shows that there are at least 2000 genes with high enrichment of H3K27me at their promoter region (Fig 8H), but only the 40% of these potential targets present a significant change at mRNA level (Fig.3B). Thus, I would like to know the opinion of the authors about these works and how they can explain the difference of their results.

We agree with this reviewer that the roles of EZH2 and PRC2 in EMT and cellular identity are highly debated, and certainly context-dependent. In fact, in Hirukawa *et al.* (2018) we show that Ezh2 inhibition impacts the metastatic capacity of Luminal B derived PDX breast cancer cell lines while having little impact on HER2 derived PDXs, demonstrating the metastatic phenotype is highly dependent on breast cancer subtype, and suggesting that Ezh2 non-canonical functions or Ezh1 function are at least partially response for the phenotype we observe.

As described in Figure 5D and revised Appendix Figure S4, the lack of EZH2 does not appear to be affecting cell cycle regulation at tumor endpoint, even with the complete loss of H3K27me3 through use of A395 and EPZ6438 inhibitor.

With regards to the H3K27me3 ChIP-Seq data (now Figure EV4F in revised), this graph shows 2000 genes total, and only a small proportion show higher enrichment of H3K27me3 at their promoter region, therefore it is likely more reflective of the targets with significant change at mRNA level (Fig. 3A-B). Overall, we agree that Ezh1 compensation is likely having a role in this model. Indeed, our ChIP-Seq indicates that some mesenchymal genes such as *Cd44*, *Vim* and *Cdh2* have significant H3K27me3 enrichment (see below), yet these genes are not upregulated in our model (Fig. 4B, Fig. 4F). Consistent with Gallardo, López-Onieva *et al.* (2024), we show that at least some of these mesenchymal genes can be upregulated with EZH2 inhibition and total H3K27me3 loss *in vitro* (revised Figure EV5D), although this doesn't prevent their gain of luminal gene expression and sensitization to tamoxifen (Fig. 7A-B, revised Figure EV5C).

We also want to point out that there are likely other epigenetic and transcriptomic regulators of mesenchymal genes in play beyond PRC2. For example, ER α can repress Snail expression indirectly (Dhasarthy *et al.*, 2009), and Snail itself is a master regulator of EMT and can repress E-cadherin and induce mesenchymal gene expression such as Vimentin (Kaufhold and Bonavida, 2014). We have therefore adjusted our manuscript text to better point out these complexities, and how the relationship between EZH2 and EZH1 warrants further investigation. We have also pointed to the use of EZH2 degraders in our revised manuscript in order to better elucidate distinct roles of EZH2 canonical vs non-canonical activity.

To clarify the role of EZH2 in epithelial-mesenchymal plasticity, I think it would be interesting to analyze how HER2+ cells respond to a more complete depletion of H3K27me3. Specifically, the derived cell lines shown in Fig.2D could be treated with EZH2 inhibitors EPZ6438 or GSK126, due to although these inhibitors display higher affinity for EZH2, they are able to inhibit EZH1 too. The authors could check by western blot if the EZH2i treatment results in a higher reduction of H3K27me3. RNA-seq would provide a comprehensive readout of the experiment to know if cells under a

complete loss of H3K27me3 preferentially up or downregulate the EMT program. Western blot of E-Cadherin, Vimentin or Snail could complement these findings, although the EMT markers are known to vary by cell type. Migration assay would be also informative to translate the gene expression changes into functionality features.

We agree that the MMTV-rtTA/EIC cell lines provided a valuable tool for us and we agree that EZH2 inhibition treatment provides a way to study this in vitro. Using EPZ6438 and EED inhibitor A395 on number of EIC derived cell lines, we confirmed by western blot and immunofluorescence (revised Appendix Fig. S4C) that H3K27me3 was totally lost upon 6d treatment. However, immunoblot analyses revealed no change in Vimentin and only a modest upregulation of E-cadherin upon EZH2i. We also could not detect other EMT markers such as N-cadherin and Snail in these cell lines. These data argue that as reviewer suggested, there are likely other EMT markers at play, which could be answered in future through RNA-Seq analyses.

Related to point 2, I suggest the authors explore how EPZ6438 and GSK126 affect epithelial-mesenchymal dynamics in additional HER2+ breast cancer cell lines such as SKBR3, HCC1954, and JIMT-1. While RNA-seq would be ideal, a targeted RT-qPCR panel of epithelial and mesenchymal markers (identified based on prior transcriptomic profiling) could provide a cost-effective alternative.

We agree with the reviewers' suggestion that additional epithelial and mesenchymal marker expression should be clarified in response to EZH2i. To address this point, we performed 10-day drug inhibition in SK-BR-3 p and SK-BR-3 LR cells, followed by a targeted qRT-PCR panel of epithelial genes *GATA3*, *FOXA1*, *NOTCH3*, *EPCAM* and *AIB1*, as well as mesenchymal genes *TP63*, *KRT5*, *SNAIL*, *CD44* and *CDH2*. Primers used have been added to Appendix Table S4. As shown in revised Figure EV5C-D, and consistent with other studies (Nie *et al.*, 2019), we see upregulation of luminal genes and observe the downregulation of *SNAIL* and *KRT5* in response to EZH2i. However, we do acknowledge that some mesenchymal genes are directly controlled by EZH2 and therefore upregulated in response to EZH2i (*CD44* and *CDH2*, Fig. EV5D), in agreement with studies pointed out by reviewer 3 (Gallardo, López-Onieva *et al.*, 2024). We have modified the text to clearly point out that the upregulation of luminal genes implies ER activation and tamoxifen sensitivity, hence still supporting the clinical relevance of our findings.

In Fig. 2D-G, the authors show that Ezh2-null cells form fewer tumors in mammary glands and lungs, and attribute this to reduced epithelial-mesenchymal plasticity. However, it would be helpful to determine whether these cells also exhibit reduced proliferation in vitro, which could independently contribute to the observed in vivo phenotype.

As indicated in Appendix Fig. S3D, we did not observe any change in proliferation *in vivo* at endpoint, and therefore we were not expecting the Ezh2-deficient tumor derived cell lines to have a proliferative defect. As reviewer #3 suggested, we clarified that Ezh2-deficient cell lines had no significant change in proliferation compared to Ezh2 wildtype cell lines, evidenced by comparable EdU incorporation and through a proliferation assay (Appendix Fig. S4A-B). Given that it has been previously shown that EZH2 inhibition can reduce cell proliferation at early stages of mammary hyperplasia (Smith *et al.*, 2019), we hypothesised that the end point tumors detected in this study reflect the ability of cells to escape dependence on Ezh2 function.

The idea of using EZH2 inhibition to sensitize HER2+ breast cancer cells to endocrine therapy is compelling. However, this effect is only shown in SKBR3 cells. The manuscript would benefit from additional validation. The authors would like to test *in vivo*, if the administration of the endocrine therapy in the MMTV-rtTA/EIC system is more effective in the EZH2fl/fl than the EZH2wt/wt background. At minimum, it would be valuable to test in the derived cell lines (Fig.2D) if ERa is induced in the Ezh2 null cells and they show higher sensitivity to Tamoxifen. Additionally, tamoxifen sensitivity assays in HCC1954 and JIMT-1 cells treated with EZH2 inhibitors would also strengthen the translational potential of these findings.

A major limitation of our MMTV-driven model is that evaluating the efficacy of endocrine based therapies is problematic as the activity of the MMTV promoter is modulated by steroid hormones such as progesterone and estrogen (Andrechek *et al.*, 2009; Cato *et al.*, 1987). Data in our lab has indeed indicated that the MMTV-rtTA/EIC wildtype system is already highly susceptible to tamoxifen *in vivo*. Further, we performed tamoxifen sensitivity assays in the derived cell lines and observed a significant growth defect in both Ezh2-proficient and Ezh2-deficient conditions. These data reflect the fact that MMTV promoter activity is directly impacted by SERMs such as tamoxifen.

In revised manuscript we provide additional validation showing that Ezh2 inhibition confers sensitivity to endocrine based therapies. To address this issue, we evaluated whether a combination of tamoxifen and Ezh2 inhibitor in HCC1954 and JIMT-1 cells would inhibit cell proliferation. Due to intrinsic elevated proliferative capacity of HCC1954, we observed that a low tamoxifen dose (2 μ M) only saw a marginal decrease in proliferation in combination with EPZ6438 (revised Appendix Fig. S6B). However, at a higher dose of 5 μ M, although tamoxifen alone produced a proliferative defect, combination of tamoxifen with GSK or EPZ was able to induce the most significant

William J. Muller
Professor of Biochemistry
Rosalind and Morris Goodman Cancer Centre
McGill University
1160 Pine avenue West
Montreal, QC, Canada H3A 1A3

McGill

proliferative defect (revised Fig. 7B). As for the JIMT-1 cells, we observed that these cells were highly sensitive to 2 μ M tamoxifen and EZH2i alone and had a complete loss of cell growth in combination (Appendix Fig. S6B). We therefore lowered the tamoxifen dose to 1 μ M where we see no effect of tamoxifen alone, but still a striking complete loss of cell growth in combination with EZH2i (Fig. 7B). Collectively we believe these data strengthen the conclusion that a combination of Ezh2 inhibitors and SERMs may be a viable approach in treatment of HER2 positive breast cancer.

In line 465, the authors described H3K27me3 levels in Ezh2 null mammary glands as "residual". In my opinion, this may be misleading, as the data suggest that approximately one-third of H3K27me3 remains unaffected. The term "remaining" or "partial" might be more accurate than "residual".

We agree with this comment and have changed the word “residual” to “remaining” as suggested.

The current title is quite attractive and suggests an important idea. However, I think it suggests a broad conclusion that may not be fully supported outside the HER2 context. I recommend specifying the subtype, e.g: "EZH2 directs HER2-enriched breast cancer progression through the modulation of epithelial cellular plasticity"

We agree with this comment and have altered the title as the reviewer has suggested.

Once again, we would like to sincerely thank our Reviewers for the insightful comments and feedback. We are confident that this strengthened, revised manuscript successfully addresses all of the concerns and is a highly relevant research article for the readership of EMBO Reports.

We thank you for your consideration and look forward to hearing from you soon.

Yours sincerely,

Professor William J. Muller, Ph.D.
Canadian Research Chair in Molecular Oncology
Professor, Department of Biochemistry, McGill University
Rosalind and Morris Goodman Cancer Institute

References

Andrechek ER, Hardy WR, Siegel PM, Rudnicki MA, Cardiff RD, Muller WJ (2000)
Amplification of the neu/erbB-2 oncogene in a mouse model of mammary tumorigenesis.
Proc Natl Acad Sci U S A. 97: 7

William J. Muller
Professor of Biochemistry
Rosalind and Morris Goodman Cancer Centre
McGill University
1160 Pine avenue West
Montreal, QC, Canada H3A 1A3

McGill

- Bernardo GM, Keri RA (2012) FOXA1: a transcription factor with parallel functions in development and cancer. *Biosci Rep* 32: 2
- Cato AC, Henderson D, Ponta H (1987) The hormone response element of the mouse mammary tumour virus DNA mediates the progestin and androgen induction of transcription in the proviral long terminal repeat region. *EMBO J.* 6: 2
- Dhasarathy A, Kajita M, Wade PA (2007) The transcription factor snail mediates epithelial to mesenchymal transitions by repression of estrogen receptor-alpha. *Mol Endocrinol* 21: 12
- Eeckhoutte J, Keeton EK, Lupien M, Krum SA, Carroll JS, Brown M (2007) Positive cross-regulatory loop ties GATA-3 to estrogen receptor alpha expression in breast cancer. *Cancer Res* 67: 13
- Gallardo A, López-Onieva L, Belmonte-Reche E, Fernández-Rengel I, Serrano-Prados A, Molina A, Sánchez-Pozo A, Landeira D (2024) EZH2 represses mesenchymal genes and upholds the epithelial state of breast carcinoma cells. *Cell Death Dis* 15: 609
- Hirukawa A, Smith HW, Zuo D, Dufour CR, Savage P, Bertos N, Johnson RM, Bui T, Bourque G, Basik M *et al* (2018) Targeting EZH2 reactivates a breast cancer subtype-specific anti-metastatic transcriptional program. *Nat Commun* 9: 2547
- Kaufhold S, Bonavida B (2014) Central role of Snail1 in the regulation of EMT and resistance in cancer: a target for therapeutic intervention. *J Exp Clin Cancer Res* 33: 62
- Liu L, Xiao B, Hirukawa A, Smith HW, Zuo D, Sanguin-Gendreau V, McCaffrey L, Nam AJ, Muller WJ (2023) Ezh2 promotes mammary tumor initiation through epigenetic regulation of the Wnt and mTORC1 signaling pathways. *Proc Natl Acad Sci U S A* 120: e2303010120
- Nie L, Wei Y, Zhang F, Hsu YH, Chan LC, Xia W, Ke B, Zhu C, Deng R, Tang J *et al* (2019) CDK2-mediated site-specific phosphorylation of EZH2 drives and maintains triple-negative breast cancer. *Nat Commun* 10: 5114
- Smith HW, Hirukawa A, Sanguin-Gendreau V, Nandi I, Dufour CR, Zuo D, Tandoc K, Leibovitch M, Singh S, Rennhack JP *et al* (2019) An ErbB2/c-Src axis links bioenergetics with PRC2 translation to drive epigenetic reprogramming and mammary tumorigenesis. *Nat Commun* 10: 2901
- Wilson BJ, Giguère V (2008) Meta-analysis of human cancer microarrays reveals GATA3 is integral to the estrogen receptor alpha pathway. *Mol Cancer.* 7: 49.
- Zimmerman SM, Lin PN, Souroullas GP (2023) Non-canonical functions of EZH2 in cancer. *Front Oncol* 13: 1233953

Dear Dr. Muller,

Thank you for the submission of your revised manuscript to our editorial offices. I have now received the reports from the two referees that I asked to re-evaluate the study, you will find below. As you will see, the referees now fully support publication of your study in EMBO reports.

Before we can proceed with formal acceptance, I have the editorial requests below I ask you to address in a final revised manuscript. Please also provide a final p-b-p-response the editorial requests.

Editorial requests:

- Please check again that the number "n" for how many independent experiments were performed, their nature (biological versus technical replicates), the bars and error bars (e.g. SEM, SD) and the test used to calculate p-values is indicated in the respective figure legends (main, EV and Appendix figures). Please also check that all the p-values are explained in the legend, and that these fit to those shown in the figure. Please provide statistical testing where applicable. Please avoid the phrase 'independent experiment' but clearly state if these were biological or technical replicates. Please also indicate (e.g. with n.s.) if testing was performed, but the differences are not significant. In case n=2, please show the data as separate datapoints without error bars and statistics. See also:

<https://link.springer.com/journal/44319/submission-guidelines#cms-Figure-and-data-presentation>

If n<5, please show single datapoints for diagrams. Moreover:

- Please note that the exact p values are not provided in the legends of figures 1B, C, D, E, F, G; 2A, B, C, D, E, F; 3C, F, G, H; 4A-F; 6D, E; 7A, B; EV1 A-C; EV2 A, B; EV3 A, EV4 C, E; EV5 A, C, D

- Please note that the box plots need to be defined in terms of minima, maxima, centre, bounds of box and whiskers, and percentile in the legends of figures 3C, EV5 A

- Please note that information related to n is missing in the legends of figures 2B, C; 3B, C, F-H; 4D

- Please note that the error bars are not defined in the legends of figures 1D, E, F, G; 2A, B, C, D, E, F; 3F-H; 4A-F; 6D, E; 7A, B; EV1 A-C; EV2 A, B; EV4 E, EV5 C, D;

- Please note that scale bar and its definition are missing for figure 2D.

- Please add scale bars of similar style and thickness to all microscopic images (main, EV and Appendix figures), using clearly visible black or white bars (depending on the background). Please place these in the lower right corner of the images themselves. Please do not write on or near the bars in the image but define the size in the respective figure legend. Presently, most scale bars are rather hard to see and have text nearby. Scale bars are missing in panel 2D (see above). Please check.

- Please make sure that all the funding information is also entered into the online submission system and that it is complete and similar to the one in the acknowledgement section of the manuscript text file. Presently, the Canada Research Chair of Molecular Oncology and James McGill distinguished Scientist award needs to be entered into the submission system (unless this one falls under McGill University (MGU) which is already entered). Please check.

- Please add the antibody and primer information (Appendix Tables S1-S5) to the Reagents & Tools Table and remove the tables from the Appendix. Please update the Appendix TOC and the call outs.

- Please confirm that for all Western blot panels (main, EV, or Appendix figures) the loading control was run on the same gel as the other proteins detected. Please note that we discourage comparisons between samples on different gels/blots, even if the samples derive from one experiment, as confounding factors reduce comparability. If unavoidable, the figure legend must state that the samples derive from the same experiment and that gels/blots were processed in parallel. If a 'representative' loading control is shown for multiple gels/blots, the intra-gel controls should be shown in the source data files, and the figure legends should describe the data displayed accurately. See our author guidelines:

<https://link.springer.com/journal/44319/submission-guidelines#cms-Figure-and-data-presentation> (section 'Electrophoretic gels and blots').

In addition, I would need from you uploaded separately:

- a short, two-sentence summary of the manuscript (not more than 35 words).

- two to four short (!) bullet points highlighting the key findings of your study (two lines each).

- a schematic summary figure as separate file that provides a sketch of the major findings (not a data image) in jpeg or tiff format (with the exact width of 550 pixels and a height of not more than 400 pixels) that can be used as a visual synopsis on our website.

Please let me know if you have questions regarding the revision.

I look forward to seeing a new revised version of your manuscript as soon as possible.

Best,

Referee #1:

The authors have adequately addressed the major and minor concerns raised in my initial review. The new EZH2 ChIP-qPCR experiments at the ESR1 locus, along with appropriate control loci, demonstrate direct EZH2 binding and resolve the main mechanistic issue. The analysis of H3K27me3 ChIP-seq data for additional luminal markers is clearly explained, and the text and model have been revised to incorporate the possibility of non-canonical EZH2 functions underlying the genes downregulated upon EZH2 loss. Figure readability has also been improved, including the labeling in Figure 5E.

The revised manuscript satisfactorily resolves all outstanding concerns and is suitable for publication in EMBO Reports.

Referee #3:

The revised version of the manuscript titled "EZH2 directs HER2+ breast cancer progression through the modulation of epithelial plasticity" by Linshan Liu et al. has addressed the comments made by this referee on the previous round, and the revisions have strengthened the manuscript. Overall, the study provides exciting and relevant findings for the field of EZH2 role in breast cancer.

I thank the authors for addressing all comments and for providing clarification. I have no further comments or suggestions. My congratulations to the authors for their study.

The authors have addressed all minor editorial requests.

Dr. William Muller
McGill University
Goodman Cancer Center
1160 Pine Ave West
Room McIntyre 507
Montreal, Quebec H3A 1A3
Canada

Dear Dr. Muller,

I am very pleased to accept your manuscript for publication in the next available issue of EMBO reports. Thank you for your contribution to our journal.

You may qualify for financial assistance for your publication charges - either via a Springer Nature fully open access agreement or an EMBO initiative. Check your eligibility: <https://link.springer.com/journal/44319/how-to-publish-with-us>

Yours sincerely,

>>> Please note that it is EMBO Reports policy for the transcript of the editorial process (containing referee reports and your response letter) to be published as an online supplement to each paper. If you do NOT want this, you will need to inform the Editorial Office via email immediately. More information is available here: <https://link.springer.com/partners/embo-press/editorial-policies#Peer%20review>